

# Internal structure of two alpine rockglaciers investigated by quasi-3D electrical resistivity imaging (ERI)

Adrian Emmert[1], Christof Kneisel[1]

[1]Institute of Geography and Geology, University of Würzburg, D-97074, Germany

*Correspondence to*: Adrian Emmert (adrian.emmert@uni-wuerzburg.de)

**Abstract.** Interactions between different formative processes are reflected in the internal structure of rockglaciers. Its detection can therefore help to enhance our understanding of landform development. For an assessment of subsurface conditions, we present an analysis of the spatial variability of active layer thickness, ground ice content and frost table topography at two different rockglacier sites in the Eastern Swiss Alps by means of quasi-3D electrical resistivity imaging (ERI). This approach

enables an extensive mapping of subsurface structures and hence the performance of a spatial overlay between site-specific surface und subsurface characteristics. At Nair rockglacier, we discovered a gradual descent of the frost table in a downslope direction and a homogenous decrease of ice content which follows the observed surface topography. This is attributed to ice formation by refreezing meltwater from an embedded snowbank or from a subsurface ice patch which reshapes the permafrost layer. The heterogeneous ground ice distribution at Uertsch rockglacier indicates that multiple processes on different time

domains were involved in rockglacier development. Resistivity values which represent frozen conditions vary within a wide range and indicate a successive formation which includes several rockglacier advances, past glacial overrides and creep processes on the rockglacier surface. In combination with the observed rockglacier topography, quasi-3D ERI enables us to delimit areas of extensive and compressive flow in close proximity. Excellent data quality was provided by a good coupling of electrodes to the ground in the pebbly material of the investigated rockglaciers. Results show the value of the quasi-3D ERI

approach but advice the application of complementary geophysical methods for interpreting the results.

## 1 Introduction

In areas of sporadic and discontinuous permafrost, subsurface conditions (e.g., thickness of the active layer, frost table topography, ice content, etc.) can be highly heterogeneous within small distances and can vary within single landform units (Schneider et al., 2013; Langston et al., 2011; Scapozza et al., 2011; Kneisel, 2010a). This is due to complex interactions

between small-scale surface conditions, topographical attributes and characteristics of the contributing area (c.f. Monnier et al., 2013; Luetschg et al., 2004; Harris and Pedersen, 1998) which influence the local ground thermal regime during landform formation (Otto et al., 2012; Lambiel and Pieracci, 2008). For a better understanding of the past and future development of periglacial landforms in areas of sporadic and discontinuous permafrost, which are particularly sensitive to climate change (Schneider et al., 2012; Harris et al., 2009), an enhanced knowledge of these interactions is needed. Therefore, the detection





and mapping of spatial variations within the internal structure can be seen as a first step towards an identification of linkages between surface and subsurface processes which are key drivers in landform development (Kääb et al., 2007; Haeberli et al., 2006). This spatial information can be valuable in combination with thermal modelling or process modelling approaches (Frehner et al., 2015; Scherler et al., 2013; Luetschg et al., 2008) which are, due to the heterogeneous subsurface conditions in

alpine environments, often reduced to one or two dimensions. Hence, a detailed knowledge of three-dimensional subsurface conditions can support the interpretation of results from such models.

Geophysical techniques such as electrical resistivity tomography (ERT), seismic refraction tomography (SRT) or ground penetrating radar (GPR) are widely used today and provide a multi-dimensional investigation of subsurface conditions in permafrost environments and corresponding landforms. For rockglaciers and similar periglacial landforms, geophysical

investigations of the internal structure can e.g., i) reveal glacier-permafrost interactions during the development of rockglaciers (Dusik et al., 2015; Krainer et al., 2012; Ribolini et al., 2010), ii) provide inferences with creep velocities (Hausmann et al., 2012; Kneisel and Kääb, 2007) and iii) enable an assessment of subsurface material composition (Schneider et al., 2013; Musil et al., 2006). In contrast to those numerous one- or two-dimensional field studies, in which vertical and lateral variations of resistivity distribution could be detected, three-dimensional investigations are sparse in the field of periglacial geomorphology.

Approaches were presented by Scapozza and Laigre (2014) and Langston et al. (2011) who graphically combined results from 2D ERT surveys to three-dimensional fence diagrams. However, these approaches do not take into account the actual three-dimensionality of the subsurface resistivity distribution and the so called '3D effects', which result from small-scale heterogeneities (Loke et al., 2013). This is achieved through the performance of a 3D inversion on a data set consisting of multiple intersecting 2D data sets ('quasi-3D' approach), which enables a horizontal mapping of resistivity variations, an

assessment of the geometry of the detected structures and consequently a more realistic characterization of three-dimensional subsurface conditions (Kneisel et al., 2014; Rödder and Kneisel, 2012).

The objective of this paper is to describe the internal structure of two morphologically different rockglaciers with regard to frost table variations/ active layer thickness (ALT), ice content and ground ice genesis. We therefore analyse the three-dimensional subsurface resistivity distribution which is modelled by quasi-3D ERI and the results from additionally performed

comparative surveys of 2D ERT and 2D SRT. Ground truthing is achieved by borehole temperature measurements. Although two-dimensional quantitative assessments of ground ice content have already been presented (Pellet et al., 2016; Hausmann et al., 2012; Hauck et al., 2011), we presume that a qualitative approach is sufficient for interpreting the results regarding rockglacier development, and as a quantitative approach would require an additional spatial assessment of porosity.

## 2 Study Sites

The investigated rockglaciers are located in the Eastern Swiss Alps, an area which is well-known for the occurrence of permafrost and numerous rockglaciers. The nearby rockglaciers Murtèl and Muragl belong to the best investigated rockglaciers (e.g., Maurer and Hauck, 2007). For the presented study, two landforms described as 'pebbly rockglaciers' according to the





relatively small grain size (dominant clast size < 20 cm) of the surface debris (Matsuoka et al., 2005) were chosen. The distance between the two sites is ca. 12 km. In September 2014, boreholes were drilled to a depth of 10 m and instrumented with thermistor strings with 15 temperature sensors at both sites. The temperature sensors are located at the ground surface and 0.2 m, 0.4 m, 0.8 m, 1.2 m, 1.6 m, 2 m, 3 m, 4 m, 5 m, 6 m, 7 m, 8 m, 9 m and 10 m below the surface. Accuracy of the installed

sensor type (Dallas, Geoprecision) is ±0.25 °C and a temporal resolution of 1h is provided.

### 2.1 Nair rockglacier

Nair rockglacier (46°31' N, 9°47' E; ca. 2845–2820 m asl; fig. 1a, c) is located at the southern slope of a small high mountain valley near the city of Celerina, Upper Engadine. It is part of a widespread rockglacier assembly at the valley slope and roots below a steep talus cone (fig. 1a). It is composed of debris material from the sedimentary rocks of the Piz Nair summit area

(mainly schist and marlstone). The Alpine Permafrost Index Map (Boeckli et al., 2012) describes the spatial distribution of permafrost in the area of the rockglacier as 'Permafrost in nearly all conditions'. The investigated rockglacier consists of several adjacent lobes, but our study concentrates on the uppermost eastern lobe which is about 90 m x 80 m in size. The occurrence of permanently frozen ground within this part of the rockglacier was reported by Ikeda and Matsuoka (2006) who performed one 2D SRT and one 2D ERT measurement (specified therein as 'NN12'). Their study describes the rockglacier as

'active', but no velocity measurements were published. A glaciation of the site during Little Ice Age (LIA) is not indicated by morphological traces but surface ice is displayed on ancient topographical maps from ca. 1917 to 1944 at the position of the present talus cones and at the root zone of the rockglacier (Coaz et al., 1925, 1946). The borehole is positioned in the upper part of the investigated rockglacier lobe.

### 2.2 Uertsch rockglacier

Uertsch rockglacier (46°36' N, 9°51' E; ca. 2570–2434 m asl; fig. 1b, d) is located at the head of a north-striking valley near the city of Bergün, ca. 2 km north of Albula Pass. It is tongue-shaped and about 500 m x 100 m in size. It has a marked topography with arcuate ridges contoured by deep furrows (vertical difference ca. 0.5 m to 3 m, fig. 1b) at the rockglacier snout. A several metres high lateral moraine at the western edge of the rockglacier next to a partly ice-filled depression (80 m x 40 m) at its proximal side indicates that an extensive glaciation existed on the rockglacier. Surface ice is displayed on a

topographical map from 1878, but pictured there only slightly larger than the extent of the recent ice patch (Coaz and Leuzinger, 1878). A similar lateral moraine is lacking at the eastern edge. Small longitudinal ridges occur in the central part of the rockglacier and several relict lobes in front of the rockglacier indicate a successive formation (fig. 1d). On the surface, isolated pioneer plants indicate inactivity. The rockglacier mostly consists of fine-grained schist from the mountain ridge between Piz Üertsch and Piz Blaisun. The rooting zone is part of a wide amphitheatre-like catchment area, where remnant ice

from a former glaciation and lateral moraines from the former glacier extent are visible (background of fig. 1b). However, this part is a little cut off from the root zone of the rockglacier. The Alpine Permafrost Index Map (Boeckli et al., 2012) describes the occurrence of permafrost at the main part of the rockglacier as 'Permafrost only in very favourable conditions'. Only the





area of the rooting zone of the rockglacier is described by the term 'Permafrost mostly in cold conditions'. The borehole is located in the lower part of the rockglacier, situated at the edge of one of the surface ridges. Some morphometrical attributes (e.g., length, slope aspect, lithology, etc.) of the rockglacier were presented by Ikeda and Matsuoka (2006) (specified therein as 'A8'), but no geophysical surveys were published.

**3 Methods**

The application of geophysical measurements for the detection of subsurface conditions is common practice in permafrost research (e.g. Hauck, 2013; Kneisel et al., 2008; Otto and Sass, 2006) and therefore only short descriptions of the basic approaches are given here. As the ranges of resistivity values for frozen and unfrozen conditions are partly overlapping, the application of complementary methods for the detection of frozen subsurface conditions is generally recommended (Schrott

and Sass, 2008; Ikeda, 2006).

**3.1 Electrical resistivity tomography (ERT)/ quasi-3D electrical resistivity imaging (ERI)**

Geoelectrical measurements are based on the varying electrical conductivity of different materials (e.g., minerals, sediments, air and water) which are assumed to be heterogeneously distributed within the subsurface. Hence, the detection of potential difference patterns can be used to derive information on the geometry of structural heterogeneities and their electrical

properties. The wide range of resistivity values for most loose materials is caused by their porosity, the varying water content and its state of matter. This connection allows to convert the image of resistivity distribution into an image of subsurface conditions. Resistivity measurements are carried out by injecting direct current into the ground via two current electrodes. Two potential electrodes are then used to measure the resulting voltage difference. The arrangement of these four electrodes is described by the 'electrode array' which determines investigation depth and sensitivity pattern of the measurement. For the

quasi-3D ERI approach (cf. Kneisel et al., 2014 for more details) data points from a 2D network of parallel and perpendicular survey lines are merged and treated as one 3D data set. Each of our 2D profile uses 36 electrodes, connected to a multi-electrode resistivity imaging system (Syscal Pro, IRIS Instruments). Further specifications of the quasi-3D ERI data sets of this study are summarized in tab. 1 and the networks of 2D lines are presented in fig. 1c, d. Dipole–Dipole (DipDip) electrode array was performed preferably due to its high resolution in the shallow subsurface and the provided time efficiency by our

multi-channel device, but also measurements with the more robust Wenner–Schlumberger (WenSl) array were performed at all sites (and partly included into the 3D data sets) to reach a higher level of reliability. Where no complete rectangular shaped grid could be set up due to topographical reasons, the model sections are partly blanked out. The measured apparent resistivity data sets were quality checked and bad datum points and outliers (standard deviation > 5%) were deleted manually. The measured 2D data sets were inverted independently for 2D interpretation and further quality checks following the procedures

proposed by Loke (2014) were performed. The 2D data sets were collated into one single 3D file using the software RES2DINVx64 (Geotomo Software). Topography was incorporated into the collated 3D data sets which were inverted using





the software RES3DINVx64 Professional (Geotomo Software). We used the robust inversion scheme (L1-norm) for the smoothness-constrained regularization, which tends to produce models with sharp boundaries (Loke and Barker, 1996). The optimization method tries to reduce the absolute difference between the calculated and the measured apparent resistivity values by adjusting the resistivity of the model blocks. These differences are quantified as a mean absolute misfit error value (abs.

Error). The inversion continues until acceptable convergence between the calculated and the measured data is reached (see Rödder and Kneisel (2012) for more details on inversion settings). For investigating model reliability, the resolution matrix approach (Loke, 2014; Wilkinson et al., 2006) was performed on all data sets. This approach provides a measure of independence of the modelled resistivity values from neighbouring cells or inversion settings. For the conversion from resistivity values to subsurface conditions, we applied qualitative attributions based on direct observations by Ikeda and

Matsuoka (2006) as their study was performed in the same area, as well as appraisals from our own borehole measurements.

### 3.2 Seismic refraction tomography (SRT)

Seismic refraction tomography (SRT) is a suitable complementary method to geoelectrical investigations as it is based on the independent parameter of seismic wave velocity (Kneisel et al., 2008). We used SRT to confirm the occurrence of frozen ground and to derive a broad threshold value to distinguish between frozen and unfrozen subsurface conditions (Seppi et al.,

2015). On each rockglacier, one SRT profile was performed. The 2D SRT survey lines were set up next to 2D ERT survey lines and in close proximity to the boreholes (fig. 1 c, d). 24 Geophones were used with an along-line separation of 3 m. We used a Geode Seismograph (Geometrics) and a sledge hammer as source of the seismic signal. Shot points were located between the geophones at Nair rockglacier and between every second geophone at Uertsch rockglacier. Data processing and analyses were performed using the software package SeisImager 2D (Geometrics, Inc.). It included a detection of the first

onset of the seismic waves on the geophones and a reciprocity check of their travel time between source and receiver location (Geometrics, 2009). A tomographic inversion scheme with an initial model based on a prior time-term inversion was used, as this method is well suited for the assumed heterogenic subsurface conditions (Schrott and Hoffmann, 2008). Surveys of comparative ERT and SRT were performed on 06 September 2015 (Nair) and 27 July 2015 (Uertsch), respectively.

## 4 Results

### 4.1 Frozen ground verification and characteristics

#### 4.1.1 Nair rockglacier

A boundary which reflects the characteristic sharp increase in both resistivity and velocity at the transition from unfrozen to frozen subsurface conditions can be obtained through the complete profiles at Nair rockglacier (fig. 2a, b). Corresponding values for this boundary are around 7 kΩm in the ERT profile and 2 km s$^{-1}$ in the SRT profile, respectively. At the position of

the borehole, the boundary reaches a depth of 4 m which is not in accordance with the depth where the temperature profile



from the day of the geophysical measurements (daily means) undercuts the 0 °C line (fig. 2c). However, values from the temperature sensors installed between depths of 3 m and 5 m vary between -0.07 °C and -0.19 °C. This means, that the difference from the freezing point is below the accuracy range of the sensors. The complete year-round temperature logging (not shown), shows that values of daily mean temperatures are consistently negative below a depth of 3 m, but only the last

two sensors of the thermistor chain show values that are consistently lower than -0.25 °C. The upslope part of the 2D profiles, where the geophysical profiles are overlapping (Y= 10–25 m), represents the steep talus cone at the root zone of the rockglacier. Active layer thickness in this part is 2 m. Resistivity values are around 4 kΩm in the unfrozen active layer and vary between 7 kΩm and 20 kΩm in the frozen layer. The latter values are considerably lower than the maximum values of the ERT model, which are in the range of several hundred kΩm in an area which is not included in the SRT profile. Only the SRT model but

not the shallower ERT model shows a second boundary in a depth of 12 m, where velocity values rise from 3.6 km s$^{-1}$ to 4.6 km s$^{-1}$. However, it must be noted that data coverage is low in this part. The downslope following part of the profiles (Y= 25–65 m) shows a steady increase of ALT from 3 m to 5 m with decreasing resistivity values from 5 kΩm to 2 kΩm. This can be linked to an increase in water content as runoff of unconfined water is reduced in this less steep part of the survey line. Resistivity values in the frozen layer are lower in this part of the model and vary between 12 kΩm and 8 kΩm. The lower

boundary, which is detected only from the SRT model, descends and disappears at Y= 36 m. The dip angle of this boundary is much steeper than the slope angle of the rockglacier surface at this position. It therefore likely represents the depth of bedrock. The permafrost layer is shaped in a slightly wavy form in both models, which shows an undulating frost table topography. The courses of this boundary are not fully synchronous between the models, but this variation can be explained by the small parallel shift between the survey lines (fig. 1c).

**4.1.2 Uertsch rockglacier**

Similarly shaped structures can be observed in the geophysical results from Uertsch rockglacier (fig. 3a, b), although the positions of the structures seem to be slightly shifted between the profiles. At the position of the borehole, results are in good agreement with the observed subsurface temperatures from the day of the measurements (fig. 3c). At a depth of 4 m, a sharp increase of resistivity and velocity can be observed. The vertical temperature profile (daily means) of the borehole from the

25   day of the geophysical surveys, reaches 0 °C at the same depth and shows that the detected boundary represents the frost table. Results from nearly year-round temperature logging at the borehole (not shown), show that down from a depth of 4 m to the end of the thermistor chain, maximum daily mean temperature values remain between 0.00 °C and -0.18 °C. This represents permanently frozen conditions, but is within the accuracy range of the sensors. In the permafrost layer of the 2D models, resistivity values are between 8 kΩm and 39 kΩm. These values can be linked to strong variations of ice content, which range

between ice-cemented and ice-supersaturated conditions (Ikeda and Matsuoka, 2006). Velocity values of this layer are between 2 km s$^{-1}$ and 3.2 km s$^{-1}$. An area of maximum high velocity and maximum high resistivity values, slightly shifted upslope in the SRT profile, is visible around the borehole location. Within the ERT profile, the detected permafrost layer ends at a depth of 11 m where resistivity values decrease again. Velocity values in the SRT section show a further increase with depth below



the thermistor chain which indicates that the frozen layer is underlain by unfrozen material with a high level of compaction. However, it must be noted that data coverage is low in this deep part of both models. Following the profile in a downslope direction, the layer of high velocity sharply descends in the SRT model between Y= 45 m and Y= 19 m. This coincides with a decrease of resistivity values to below 5 kΩm and therefore likely represents unfrozen conditions over the complete depth in this part of the profile. The position of this unfrozen part corresponds to the end of the surface ridge structure where the borehole is placed on. In the further downslope part of the profile, the detected structures resemble again those from the upslope part. Near the surface, low resistivity and low velocity values indicate an ALT of 6 m which is higher than the ALT of the upslope part.

## 4.2 3D subsurface models

### 4.2.1 Nair rockglacier

The 3D model of subsurface resistivity distribution at Nair rockglacier (fig. 4) shows a strong and stepwise decrease of resistivity values in Y-direction. The range of modelled resistivity values spans from 420 kΩm in the part of the model which corresponds to the talus cone (cf. fig. 4, first slice) to <1 kΩm in the shallow subsurface of the downslope part of the model. Variations in X-direction are less pronounced and only show a slight increase of resistivity values from the margin of the rockglacier towards the centre of the rockglacier. The highest resistivity values aggregate in a 15 m to 25 m long and ca. 40 m wide structure with values between 200 kΩm and 400 kΩm in its central part. These values indicate highly ice-supersaturated debris material (cf. Ikeda and Matsuoka, 2006). It is overlain by a 2m thick layer of lower resistivity values, which vary between 4 kΩm and 8 kΩm and is therefore regarded as the unfrozen active layer. The structure of high resistivity is not shaped homogeneously but narrows slightly along the X-axis towards the blanked out part from Y= 42 m to Y= 30 m. The vertical extent of the structure exceeds the maximum depth of the model (15 m). In a downslope direction, at the transition between talus cone and rockglacier, the structure of very high resistivity values ends with a sharp drop of resistivity to values which vary between 20 kΩm and 12 kΩm. These values, although over one magnitude lower than the maximum values, also represent frozen material as confirmed by the 2D models, but indicate a different type of ice genesis. The upper layer with lower resistivity values increases in thickness in a downslope direction and towards the margin of the rockglacier (up to 4 m thickness), while it remains thin in the centre of the rockglacier (2 m to 3 m thickness). This forms a continuous and convex shaped frost table. Resistivity values in the upper layers are also reduced in this part of the model and vary between 4 kΩm and 6 kΩm, which corresponds to similar observations from the 2D models. The adjacent part, which corresponds to the main part of the rockglacier, generally shows much lower resistivity values. Like the upslope parts of the model, it can be divided vertically into two layers. An upper layer with values from 1 kΩm to 3 kΩm (unfrozen, high liquid water content) can be delimited from deeper parts of the model where resistivity values vary between 7 kΩm and 16 kΩm (frozen, probably ice-cemented). In Y-direction, the boundary between the two layers is descending stepwise to a vertical difference of 4 m. This forms a wedge-shaped outline of the permafrost layer which penetrates into the rockglacier from the talus cone. Behind Y= 85




m, the deeper layer disappears, except for a few cells at the outer margin of the resistivity model which are considered as artefacts from the inversion process. This part of the model represents a downslope thickening of the active layer and a decrease in ice content from ice-cemented to ice-free and hence probably permafrost-free conditions. The stepwise increasing ALT leads to an undulating topography of the frost table which indicates a successive formation. At the eastern part of the model,

in front of the blanked out part, structures of high resistivity values do not form a consistent layer but display a patchy distribution. Resistivity variations in X-direction show that the ice content also decreases from the central part of the rockglacier towards the eastern margin, which is accompanied by an increase of ALT. This part of the model corresponds to a part of the rockglacier which is positioned below a rockwall and not below the talus cone.

### 4.2.2 Uertsch rockglacier

The resistivity model of the lower part of Uertsch rockglacier (Uertsch_01, fig. 5) shows a heterogeneous pattern of small units. Modelled resistivity values vary between <0.1 kΩm and ca. 40 kΩm. In the western part of the model, a 10m wide curved structure of extraordinary low resistivity values (up to 1.5 kΩm) is visible. This structure remains consistent through the complete model depth of 15 m and represents unfrozen subsurface conditions. It corresponds to a ridge at the western margin of the rockglacier which is ca. 1 m to 2 m high in this part and a continuation of the lateral moraine in the upslope part

of the rockglacier. The areas of the model which show high resistivity values can be divided into several tongue-shaped structures which root in a common zone (marked as (1) in fig. 5). These structures can be delimited by extrapolating the threshold value for frozen and unfrozen conditions which was gained from the comparison of geophysical and temperature data on the whole rockglacier, which is rather homogenous regarding debris size composition. Down to a depth of 6 m, three of those structures are visible at X= 14 m (2), X= 24 m (3) and X= 46 m (4a, b). They are outlined by a band of lower resistivity

values. Below a model depth of 6 m, the boundaries between the tongue-shaped structures vanish, but some parts of the high resistivity structures remain. The first of the three structures is 55 m long and reaches a depth of 7 m. Resistivity values vary between 7 kΩm and 14 kΩm. The layer above this structure is highly variable in thickness and ranges from 4 m in the upslope part of the model over 1 m in the middle part to 5 m in the downslope part of the model. The second structure is about 70 m long and 10 m wide. This structure is hit by the borehole as marked in fig. 5. High resistivity values of up to 30 kΩm aggregate

in a layer of 6 m to 8 m thickness. The structure occurs in a depth of <1 m to 2 m and dips in a downslope direction. A shallow covering layer of low resistivity values exists only partly and shows values of up to 1.5 kΩm. Below the layer of high resistivity, values decrease again to around 1 kΩm to 3 kΩm. About the same values are reached in the front of the structure, where a prominent U-shaped patch of lower resistivity values is visible (fig. 5, depth slices 3–4 m, 4–5 m,). The third area of high resistivity values is 70 m long and reaches to the northern boundary of the model. The upslope part of the structure (4a) is

visible down from the first model slice while the downslope part (4b) initially occurs in a depth of 3 m underneath a layer of resistivity values of around 5 kΩm. A lower boundary of the structure could not be delimited within the model boundaries in the upslope part, while the downslope part can be detected down to a depth of 9 m. The resistivity distribution clearly reflects the arcuate ridges of the surface topography and illustrates frozen conditions within the broad ridges and unfrozen conditions



below the interrupting furrows and below a small surface depression (corresponding to U-shaped patch of lower resistivity values). The investigated part of the rockglacier lacks a continuous frost table and the strongly varying ALT indicates a disturbed development. It likely reflects a disequilibrium between the modern environmental conditions and those at the time of ground ice formation.

The resistivity model of the upper part of Uertsch rockglacier (Uertsch_02, fig. 6) directly follows the Uertsch_01 grid in an upslope direction (cf. fig. 1d). It shows a wider range of resistivity values (min: <0.1 kΩm; max: 250 kΩm) and differences in the resistivity distribution pattern. One dominant high-resistivity anomaly occurs at the central western side of the model (5). It reaches resistivity values from 35 kΩm to 250 kΩm. While the upslope part of the structure partly occurs directly below the surface in the first model slice, the downslope part is only visible down from a depth of 2 m where the structure reaches its

maximum spatial extent. It remains spatially constant down to a depth of 5 m, where a slight decrease in resistivity values can be observed. The decrease in extent and resistivity of the structure increases with depth and at the lower boundary of the model (20 m) values of below 4 kΩm are displayed. The position of this structure corresponds to a surface depression which was partly covered with snow and ice at the date of the 3D survey. Resistivity values indicate highly ice-supersaturated conditions and/ or altered ice of sedimentary origin. The downslope increasing ALT points towards an incorporation of an ice patch. A

second, rather undefined pattern of high resistivity values (6a, b) becomes apparent as a spatial structure below a depth of 2 m, although some model cells in the area already display high resistivity values in upper depth slices. Attaching this structure to the resistivity distribution of Uertsch_01, reveals that this structure forms a precursor of the three tongue-shaped high resistivity structures in the lower part of the rockglacier. In the shallow subsurface of Uertsch_02 model, the structure displays values from 7 kΩm to 18 kΩm which represents frozen conditions. In a depth of 6 m, it loses an elongated segment (6a) and

is reduced in extent to an area of 14 m x 30 m. It reduces further downwards by extent and by resistivity and is detectable only down to a depth of 11 m. Its connection to the central part of the high resistivity structures of Uertsch_01 model indicates that it represents an upslope part of the arcuate ridge structures. On the rockglacier surface, the resistivity structure corresponds to an area of longitudinal ridges with 0.3 m to 0.5 m height. Like in the downslope part of the rockglacier a continuous frost table does not exist, but ALT variations are less pronounced.

## 25   5 Discussion

### 5.1 Methodological aspects

The quality of quasi-3D ERI models is influenced by the separation of the survey grid lines. In our investigation, the chosen value of twice the along-line electrode separation as distance between the Y-lines of all survey grids provides an adequate data coverage for investigations of the shallow subsurface (Gharibi and Bentley, 2005). The separation factor of X-lines was

adjusted due to site–specific reasons, like deep snow fields or topographical obstacles and ranges between a factor of 3 and 6 of the along-line electrode spacing. However, this is still assumed to be sufficient as the application of perpendicular grid lines is not a mandatory requirement for 3D data acquisition but useful for the delimitation of small-scale structures orthogonal to





the survey line direction (Loke et al., 2013; Rödder and Kneisel, 2012; Chambers et al., 2002). Despite the known limitations of the Dipole–Dipole electrode array under rough surface conditions (rather low signal strength), its application as a basic electrode array in this study provided very good results. This was likely promoted by the pebbly debris material which improved the coupling of the electrodes to the ground. The suitability of the approach is shown by the excellent data quality

(only 4% of all data points were removed, cf. tab. 1) and by a comparison between the results from the independently inverted Dipole–Dipole data points and Wenner–Schlumberger data points of the Uertsch_02 data set, which provided similar results (not shown). Additionally, we observed only a slight increase in misfit errors for the 3D models compared to the independently inverted 2D models, which indicates a precise grid setup. The resolution matrix approach shows, by using a cut-off value of > 0.05 (Hilbich et al., 2009; Stummer et al., 2004), that most parts of the 3D models are significantly resolved (not shown). Only

those parts of the model layers which are underneath the high resistivity anomalies and furthermore deeper than 9 m are weakly resolved and are hence interpreted carefully. Results from the comparative 2D SRT/ 2D ERT surveys show good structural accordance and validate the chosen approach. The detected structures from our geoelectrical investigations of Nair rockglacier broadly resemble the structures which were detected by Ikeda and Matsuoka (2006), who performed a single 2D ERT survey on the same rockglacier. The velocity values from their study for a two-layered subsurface (0.34 km s$^{-1}$; 2.9 km s$^{-1}$) were also

broadly reproduced in our study, but they delimited a 2.2 m to 2.4 m thick active layer which we could not rediscover in our study around a decade later.

## 5.2 Range of resistivity values

For most parts of the presented rockglacier models, the range of resistivity values for frozen conditions is rather low compared to other rockglacier sites (Seppi et al., 2015; Dusik et al., 2015; Kneisel, 2010b; Hilbich et al., 2009; Maurer and Hauck, 2007)

and is closer to those of landforms in fine-grained environments (Lewkowicz et al., 2011; Farbrot et al., 2007; Kneisel et al., 2007; Ross et al., 2007). Nevertheless, borehole temperature measurements and comparative SRT surveys presented in this study indicate the occurrence of permafrost, although velocity values are in the lowermost range for frozen ground (cf. compilation of Draebing (2016)). The detected threshold values for frozen condition of around 7 kΩm at Nair rockglacier and around 8 kΩm at Uertsch rockglacier are plausible for the sites as Ikeda and Matsuoka (2006) found similar values for frozen

conditions at a similar rockglacier by a comparison between direct observations in a pit and ERT. However, the extrapolation of single threshold value on a whole rockglacier can be problematic due to variations of grain size and porosity. We think that the extrapolation is suitable at the investigated sites, as such variation are not visible on the surface. The low level of resistivity values at the investigated sites can be explained by the small grain size of the debris material which is known to show much lower resistivity values in a frozen state than ice-bearing bouldery materials. This is caused by the lower pore space volume

and hence the lower ground ice volume which can develop by freezing of unconfined water (Scapozza et al., 2011; Vonder Mühll et al., 2000). Additionally, at temperatures only slightly below the freezing point, the fine grained debris material can preserve a relatively high amount of liquid water even at sub-zero temperatures, which can cause the observed low resistivity values (Schneider et al., 2013). The non-crystalline origin of the talus material can be listed as another factor which influences



the local resistivity regime (Etzelmüller et al., 2006). A high proportion of unfrozen water can also be a reason for the mismatch between the observed frost table depths in the vertical plots of resistivity, velocity and temperature at Nair rockglacier (fig. 2c). As the geophysical approaches are affected by material properties and not directly by temperature, the characteristic increase of resistivity and velocity does not coincide with the depth of the observed sub-zero temperatures (Pogliotti et al.,

2015; Hauck, 2002). However, sensor accuracy must also be taken into account. Results from ERT and SRT at Uertsch borehole do not show this mismatch phenomenon. This might be linked to an unhindered drain of unconfined water into the adjacent unfrozen part of the profile.

## 5.3 Structure of Nair rockglacier

Ice content distribution at Nair rockglacier is probably determined by an ice patch of sedimentary origin, buried under the

steep talus cone. This is indicated by resistivity values of up to 420 kΩm which display highly ice-supersaturated conditions. The occurrence of such structures in the root zone of rockglaciers is a well-known phenomenon which develops from buried snowbanks or ice patches which were incorporated into the subsurface, e.g. by rockfall (Monnier et al., 2011; Lugon et al., 2004; Isaksen et al., 2000; Haeberli and Vonder Mühll, 1996). A high geomorphological activity is present at Nair rockglacier where frequent rockfall events were observed during several field campaigns. The sharp drop in resistivity values to 20 kΩm

shows that a clear distinction between the embedded snowbank and the adjacent part is present. This may indicate a relatively young age of incorporation on time scales of landform development, as this contrast would probably diminish through time. A qualitatively similar drop of resistivity values was observed by Ribolini et al. (2010) at Schiantala rockglacier (French Maritime Alps), where it separates an area of debris-covered sedimentary ice and an area of typical permafrost ice at the margin of an LIA glaciation. However, linking the incorporation of sedimentary ice into Nair rockglacier to the end of LIA, for which

a glaciation is displayed on ancient maps (Coaz and Leuzinger, 1878), is speculative. The occurrence of ice of sedimentary origin below the talus cone is also an adequate interpretation for the downslope following part of Nair rockglacier which shows resistivity values typical for congelation ice and a decrease from 20 kΩm to 12 kΩm. This represents a gradual decrease of ice content and is associated with a stepwise increase of ALT. The formation of this wedge-shaped structure can originate from meltwater of the embedded ice which drains into the rockglacier during summer and refreezes (Isaksen et al., 2000). The ability

of the pebbly material to store water and to reduce the speed of runoff is likely to support this process (Ikeda et al., 2008). It also explains the lower ice content at the lateral margin of the rockglacier which would also be less affected by meltwater flow. If we consider the described processes as a current phenomenon, the ice-free parts at Nair rockglacier are equal to the permafrost-free parts and vice versa. Stable ground ice conditions at the rockglacier are indicated by the gradual shape of the continuous frost table, the generally permafrost-favourable conditions according to the Alpine Permafrost Index Map (Boeckli

et al., 2012) and own nearly year-round measurements of ground surface temperatures (not shown).



## 5.4 Structure of Uertsch rockglacier

The observed ice content variations in the lower part of Uertsch rockglacier indicate that different processes were involved in rockglacier development. The ice-free and hence probably unfrozen ridge at the western edge of the rockglacier (extension of the lateral moraine) seems to be unconnected with the more differentiated central part, where the difference between frozen arcuate ridges and unfrozen furrows contrasts the commonly known distribution, which is usually attributed to topographic or microclimatic effects (Hanson and Hölzle, 2004; Harris and Pedersen, 1998). The observed ice-free lateral moraine and the surface ice patch suggest that interactions between glacial and periglacial processes occurred, like they were assumed for multiple other rockglaciers (Monnier et al., 2011; Ribolini et al., 2010; Berger et al., 2004; Lugon et al., 2004). Glaciation during LIA as illustrated on an ancient topographical map from the 1870s, was only slightly more extensive than the recent surface ice patch and remained upslope the today ridge-affected part (Coaz and Leuzinger, 1878). But, like Monnier et al. (2013) pointed out for Sachette rockglacier (French Alps), one of the several other glacier advances during Holocene could have overridden the rockglacier. The occurrence of buried ice of sedimentary origin is conceivable only for the well-defined structure in the central western part of Uertsch rockglacier (Uertsch_02 model), which corresponds to the presumably glacial depression. Next to a surface snow field which existed during the days of the 3D survey within this depression, maximum resistivity values were detected (250 kΩm). Although this is still not in the range of sedimentary ice which typically reaches up to several MΩm (Haeberli and Vonder Mühll, 1996), a subsurface formation of congelation ice by refreezing meltwater from sedimentary surface ice or an alteration of remnant sedimentary ice by multiple freeze-thaw cycles seems possible, especially in case of a former glaciation. The occurrence of buried ice of glacial origin within the rockglacier snout, as also frequently observed at other sites, can be excluded by the range of the modelled resistivity values (< 70 kΩm), which rather indicates distinct patches of congelation ice within the ridges by the three tongue-shaped structures of high resistivity values. A similarly shaped elongated geometry of the frozen structures was mapped in the Dolomites by Seppi et al. (2015). The formation of the characteristic ridge-furrow topography on the rockglacier surface is known to result from compressive processes (Frehner et al., 2015; Springman et al., 2012; Kääb and Weber, 2004). At Uertsch rockglacier, the model of overthrusting lobes, as generally presented by Kääb and Weber (2004) fits well to the observations. Ridge formation is assumed to induce a local enrichment of ice content through a thickening of ice-saturated layers (Ikeda and Matsuoka, 2006) and can therefore explain the relatively high ice content within the transverse parts of the arcuate ridges in contrast to the lower ice content of the upslope longitudinal parts. A patchy occurrence of relatively high ice content within the ridges may also explain the low variation of borehole temperatures around the freezing point as much thermal energy will likely be lost by phase transitions of water. A thickening of the active layer as described by Haeberli and Vonder Mühll (1996) could only be detected in the lowermost part of the ridges where the deformation is strongest. The upslope central part of the rockglacier, where ridges are shaped in longitudinal direction and show resistivity values from 8 kΩm to 20 kΩm lacks this thickening, likely due to a reduced dynamic forming. Regarding again the model of overthrusting processes, Kääb and Weber (2004) presumed that those lateral ridges are the result of ridge deformation by a decrease in speed at the margins of the flow area. This former marginal

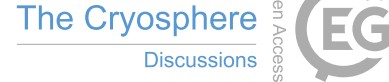

position of the longitudinal ridges supports the concept of a successive rockglacier development. We assume that ridge formation at Uertsch rockglacier is connected to tongue-shaped areas of creep activity on the rockglacier surface. The lower resistivity values of the extensive flow structures are likely caused by freezing of ionically enriched groundwater at the permafrost base (Haeberli and Vonder Mühll, 1996). However, a glacial formation of the ridges, as e.g. observed by Monnier

et al. (2011) at Thabor rockglacier cannot be excluded as a general influence of surface ice on the formation of Uertsch rockglacier is obvious.

## 6 Conclusions

The application of quasi-3D ERI enables the detection and mapping of permafrost conditions in a spatially extensive way. At Uertsch rockglacier, the approach showed its value for the delimitation of several small-scale frozen structures within

heterogeneous subsurface conditions. The rather homogenous subsurface layering at Nair rockglacier excludes the occurrence of major structural anomalies and shows an undulating frost table topography as well as a gradual decreasing ice content. An excellent data quality was promoted by the pebbly grain size of the investigated rockglaciers and permitted the extensive application of Dipole-Dipole electrode array. However, due to the specific conditions at pebbly investigation sites, concerning e.g. the range of resistivity values and the influence of grain size on the occurring processes, results should be extrapolated to

bouldery rockglaciers only carefully. Inversion characteristics and additionally performed comparative surveys indicate reliable results and emphasize the suitability of the approach. Our results show that the following subsurface characteristics and their small-scale spatial variations can be derived from quasi-3D ERI and interpreted in combination with geomorphological observations from the investigated rockglaciers:

- Buried ice of sedimentary origin is a crucial factor for rockglacier development as related processes (e.g. meltwater flow) can influence ice content distribution and reflect past glacier–permafrost interactions.
- Mapping frost table topography and consistency allows to infer the currency of shaping processes like melt–freeze cycles, and hence displays a state of equilibrium or disequilibrium to modern environmental conditions.
- Quasi-3D ERI results in combination with the observed surface ridge structures show that areas of compressional and

extensional flow occur in close proximity and indicate a successive rockglacier development.
- Meltwater can strongly increase ground ice content in downslope and less inclined parts of rockglaciers and changes its distribution to a gradual decrease which follows the observed rockglacier topography and results in a wedge-shaped outline.

To further improve our understanding of landform development, the additional application of GPR and the setup of a denser

network of SRT profiles are advised as these approaches are preferably used for the detection of layer boundaries. The presented resistivity mapping further allows an overlay of resistivity distribution with mapping of surface velocity and subsurface porosity. This might be a next step towards an identification of surface–subsurface process interactions.





**Acknowledgements**

The authors gratefully acknowledge the German Research Foundation (DFG) for financial support (KN542/13-1). Thanks to Danilo Fries and Carina Selbach for their assistance in the field and Christian Hauck for constructive comments.

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





**Table 1: Details on quasi-3D ERI surveys.**

|  | NAIR | UERTSCH_01 | UERTSCH_02 |
| --- | --- | --- | --- |
| Acquisition dates | 12–14 Aug 2014 | 11–13 Sept 2014 | 26–31 Jul 2015 |
| Number of grid lines | 17 (9 X, 8 Y) | 17 (9 X, 8 Y) | 30 (17 X, 13 Y) |
| Outline of investigated area | 70 m x 105 m | 70 m x 105 m | 74 m x 105 m |
| Number of collated 2D profiles | 17 | 20 | 60 |
| Array types for 2D profiles | 17 DipDip | 17 DipDip + 3 WenSl | 30 DipDip + 30 WenSl |
| Number of data points for inversion | 6294 (of max. 6936) | 7372 (of max 7800) | 20523 (of max. 20880) |
| Abs. Error (Iteration) | 7.77 (5) | 5.85 (5) | 8.86 (5) |



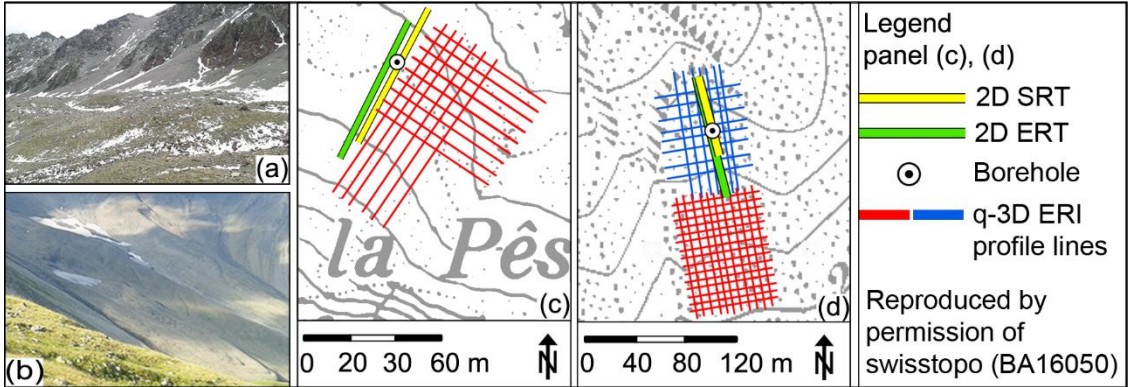

**Fig. 1: Site overview and measurement setups. a) Photo Nair site b) Photo Uertsch site c) quasi-3D ERI setup Nair d) quasi-3D ERI setups Uertsch_01 (blue lines) and Uertsch_02 (red lines).**





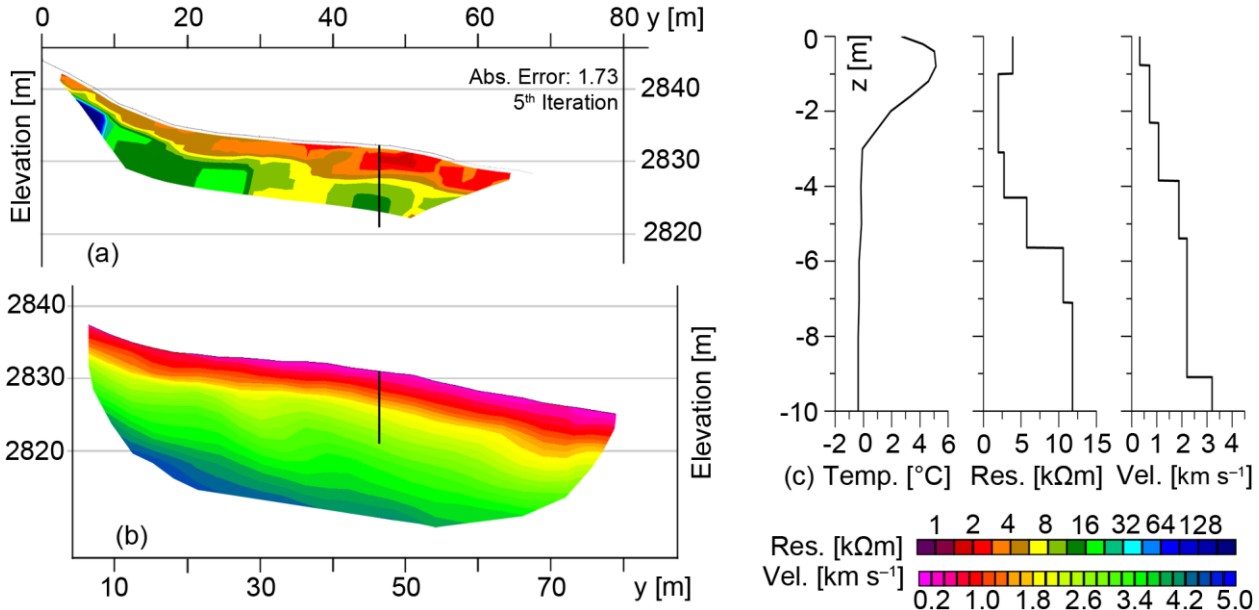

Fig. 2: Comparative analysis of (a) 2D ERT Profile, (b) 2D SRT Profile and (c) 1D Temperature/ Resistivity/ Velocity plots at Nair rockglacier.



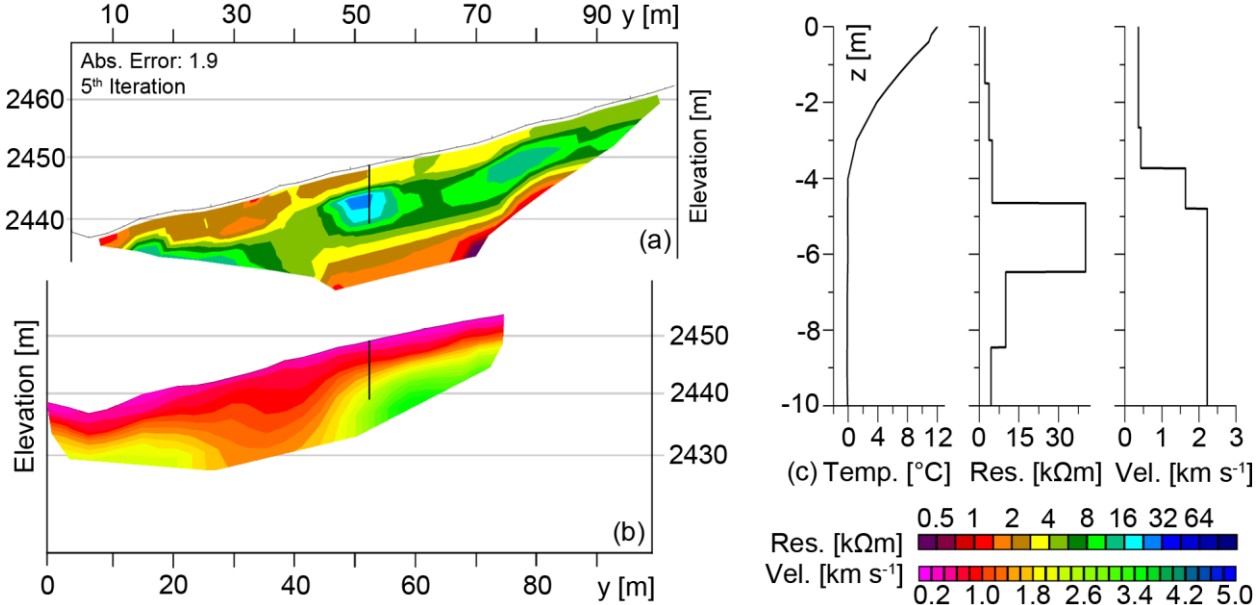

**Fig. 3: Comparative analysis of (a) 2D ERT Profile, (b) 2D SRT Profile and (c) 1D Temperature/ Resistivity/ Velocity plots at Uertsch rockglacier.**



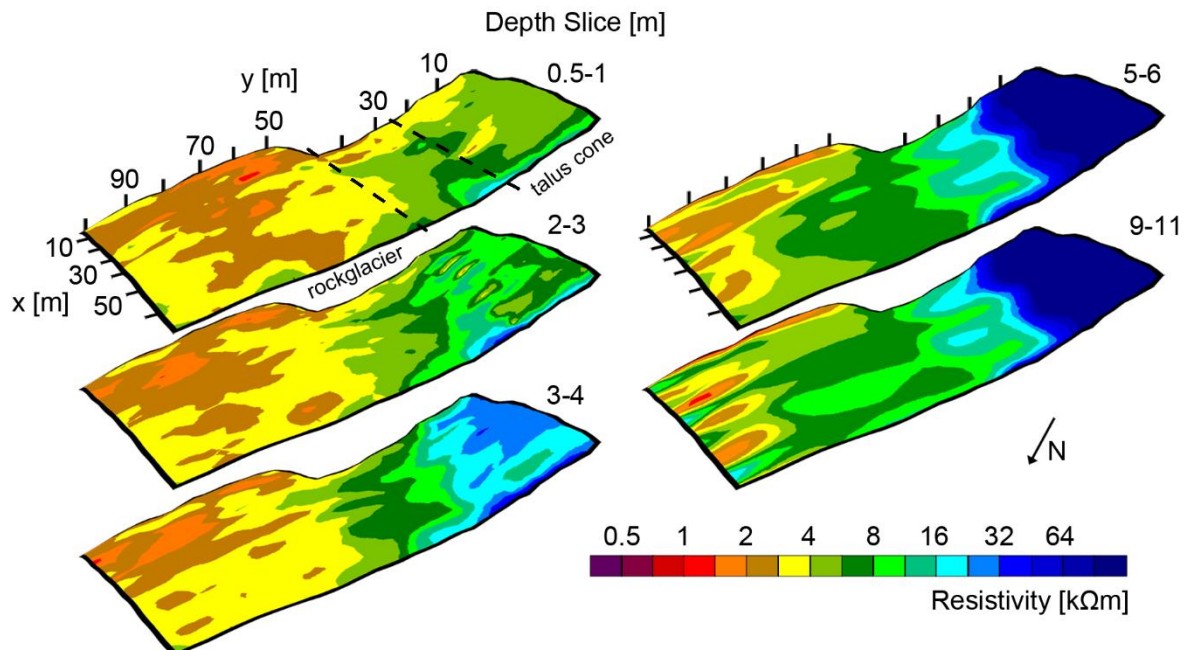

Fig. 4: Quasi-3D ERI Model Nair. Chosen depth slices.





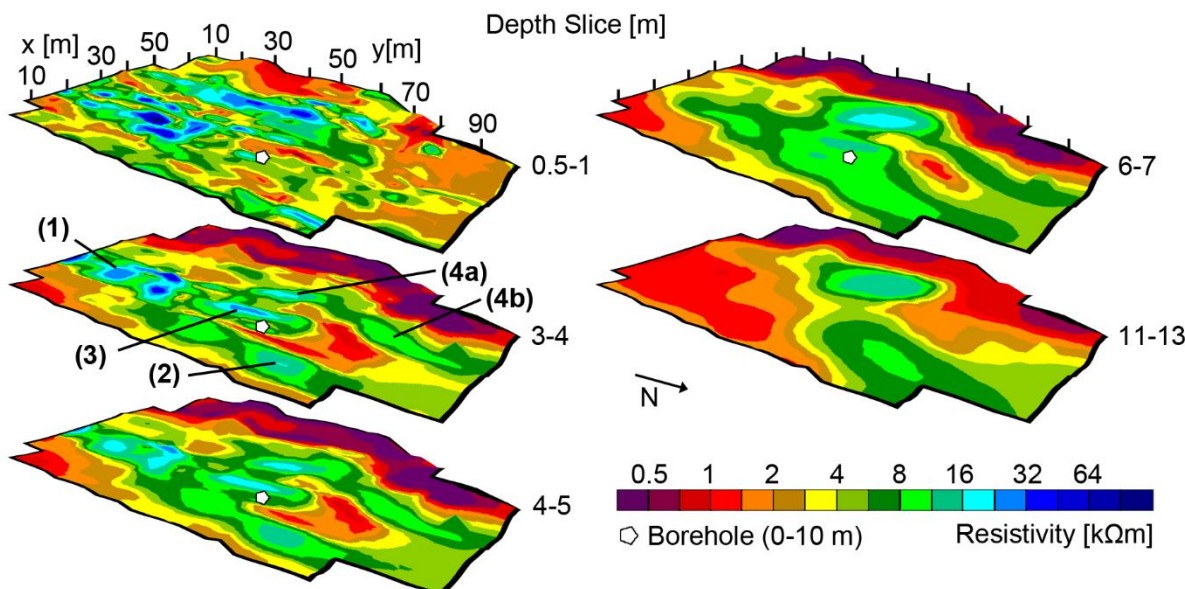

**Fig. 5:** Quasi-3D ERI Model Uertsch_01. Numbers refer to structures described in the text. Chosen depth slices.





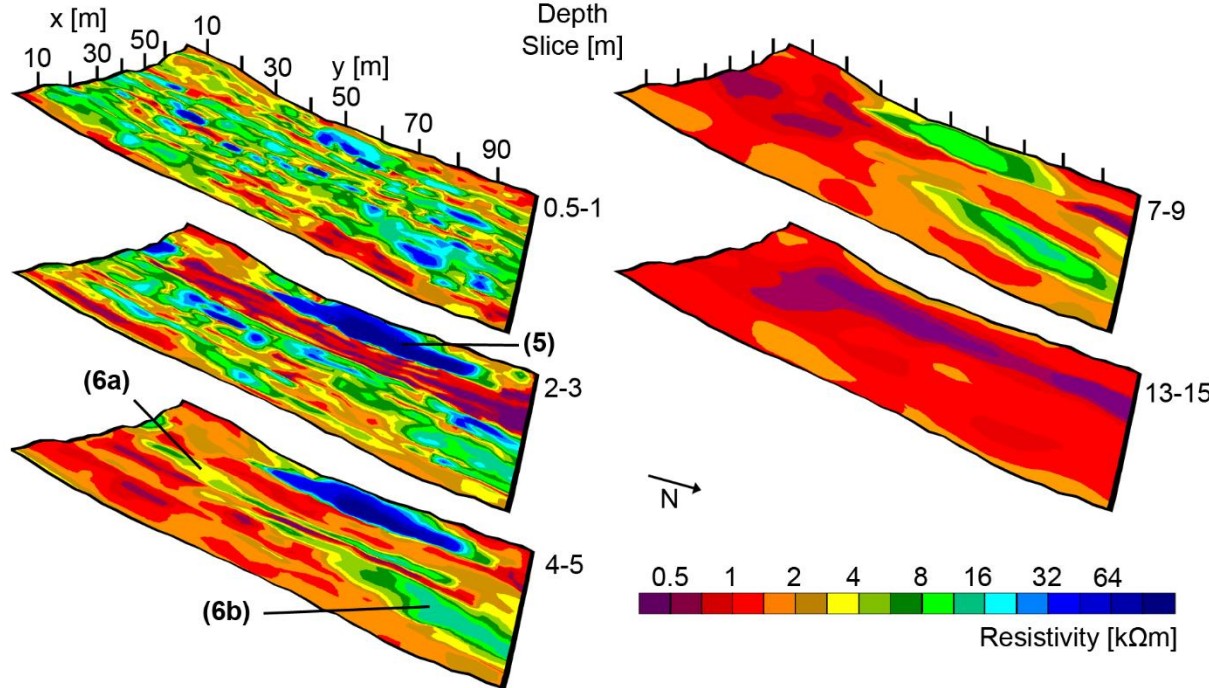

Fig. 6: Quasi-3D ERI Model Uertsch_02. Numbers refer to structures described in the text. Chosen depth slices.