# Peer review of "Internal structure of two alpine rockglaciers investigated by quasi-3D electrical resistivity imaging"

_The Cryosphere, 2016_

## Referee Comment (RC1) · Anonymous Referee #1 · 7 Sep 2016

General

Even though a number of 2D geophysical surveys have been done on many rock glaciers, newly revealed subsurface heterogeneity is still valuable for discussing the kinematics and thermal history of rock glaciers. In particular, this paper shows detailed subsurface heterogeneity through a quasi-3D method. In addition to presenting valuable data, the authors well reviewed the related articles and carefully discussed the data. However, readers probably feel it difficult to follow the discussion on surface topography, because the visible information such as photos and maps are quite limited in this paper. To improve the paper, I recommend adding some geomorphological maps.

[Figure]

Please also note the supplement to this comment:
http://www.the-cryosphere-discuss.net/tc-2016-135/tc-2016-135-RC1-supplement.pdf

[Figure]

**Supplement:**

Minor remarks

p.1

l. 25    topographical attributes and characteristics of the contributing area

It is difficult to understand what these terms mean.

l. 27 to l. 2 (p. 2)

The sentences connecting with "Therefore" refer to different topics, thus the connection seems to be logically wrong.

p. 2

ll. 10-12

It would be better to revise the sentence, because the content of iii) is not equivalent to those of i) and ii). In general, iii) is firstly interpreted, then i) and ii) are discussed.

p. 3

It would be better to add a comment whether direct observation of the borehole core is available or not. If available, it would be much better to discuss the composition of the rock glaciers with the core stratigraphy

l. 14    2D SRT

This is incorrect. They only did conventional refraction survey.

ll. 21-31

Readers cannot check the topography mentioned in this part, because the photo and map in Fig. 1 are too small. Add a geomorphological map indicating not only the survey lines but also the lateral moraine, ice patches, rooting zone, ridges/furrows and LIA glacier extent.

p. 7

l. 13    first slice

Which is the first slice in the figure?

l. 25    the centre of the rockglacier

It may confuse readers, because the center of the rock glacier does not corresponds to the center of Fig. 4 but the lower edge.

p. 8

l. 21    reaches a depth of 7 m

Is it correct? Fig. 5 indicates that the deeper part (11-13 m) also has similar resistivity.

ll. 22-23    The layer above this structure is highly variable in thickness and ranges from 4 m in the upslope part of the model over 1 m in the middle part to 5 m in the downslope part of the model.

It is impossible to understand where "this structure" is, because the relatively high resistivity structure (>6 kohmm) indicated as (2) is covered with a thin low resistivity layer only seen in

the slice of 0.5-1 m.

l. 32 to l. 1 (p. 9)

Readers cannot confirm the relation between surface topography and resistivity from Figs. 5 and 6. Add a hillshade image with contour lines of the survey area.

p. 10

ll. 29-30    This is caused by the lower pore space volume and hence the lower ground ice volume

This interpretation is questionable. Reconsider basic permafrost physics that frozen finer soil holds more unfrozen water in it near the melting point.

p. 11

ll. 22-26

I cannot understand what the authors want to say.

p. 12

The paragraph of the section 5.4 is too long. It would be better to revise it into several paragraphs.

ll. 4-6    , where ... 1998)

It would be better to remove this part from this sentence, because the topic is not discussed here but after the glacier-permafrost interaction.

---

## Referee Comment (RC2) · Anonymous Referee #2 · 27 Sep 2016

This manuscript describes the field investigation of the internal structure of two alpine rockglaciers using quasi-3D ERI. Also, this study compares these results with 2D SRT and ERT transects, and with thermistor chain data. The subject of this study is of great interest for the Cryosphere community. I enjoyed reading it. Although I recommend this manuscript for publication, I suggest some moderate revisions.

Major Comments:

1) The use of quasi-3D inversion is great and very valuable. However, I think its value cannot be really evaluated here because Figure 4 is not sufficient to analyze results. I suggest that the author also show a fence-type figure displaying various transects in 3D. This would help to see the variations in both depth and elevations. Also comparing

results from a few transect inverted separately vs together would be useful to help the reader understand the value of quasi-3D ERI.

2) The description of the Figure 4 (5 and 6) and discussion in the manuscript is great (including the discussion of past studies). However, it is very hard to follow the discussion given the limited information in the figures. Topography is likely a key property in such analysis and has been measured (as visible in Figure 4) (I suggest that the authors add a sentence to explain how they measured it in more details). It would be great to have a map of the topography so that below structure could be more easily compared to surface variability (topography and micro-topography). Figure 4 shows that topography has been included in inversion but currently it is hard to observe and compare surface and subsurface features. Also a geomorphological map or any other information about the surface properties could significantly strengthen the paper.

3) I agree that the interpretation of thermistor data, EMI and SRT is generally difficult at temperature around 0 C. I think it would be very useful to see temperature profiles at other times of the year too for enhanced understanding of where the system is expected to be more dynamic. Please consider adding such a dataset in a figure.

4) There are several locations where wording could be possibly improved. E.g., "Its" or "their" on Line 6 (what does the "its" refer to) ?; "and hence the performance"(Line 10) sounds awkward; "homogeneous decrease" or "constant decrease" (Line 12) ?; etc.

Minor Comments:

L. 28 on page 2: Consider rewording last sentence.

L.31 on page 2: "best investigated" in the world ? Please be more specific

L.2 on page 3: "boreholes" at each site ? one at each site ? please clarify. I also suggest that authors name the two sites in this section.

L.21, page 4: what is the inter-electrode spacing (2m ?)

L.28, page 4: standard deviation larger than 5%. Please provide information on how this standard deviation is measured, by the acquisition system alone ? or using reciprocal measurements ?

L.30 page 4: "collated". Consider using "merged"

L. 17, page 5: the shots were all located on the transects. Please discuss shortly why no shot have been performed outside the geophone interval (as often done in seismic refraction).

Section 4.1: I got lost in this section because I did not understand what was discussed first. I suggest that the author clearly state that this section compare single ERT transect and SRT. I think personally that this section should be placed after the quasi-3D inversion and be only related to comparison between seismic and ERT.

Line 5, page 6: "last two sensors". At which depths ? The sentence is confusing, although I agree that temperature close to 0 C means that comparison between various dataset is difficult, as well as the assessment of permafrost presence or not.

Line 5 on page 10: "removed" using which criteria

Line 9 on page 10: 'cut-off' value. Please clarify process. Although it is hard to evaluate penetration depth without knowing the acquisition geometry (a and n spacings), it seems to me that the bottom part in Figure 3 is not necessarily real. Please clarify your thoughts about this zone in the manuscript.

L.15, p.9: "could not rediscover" or "did not rediscover (or observe)". Please clarify the potential reason. Is it because environmental/climate change, thermal state during survey, difference in acquisition geometry ?

L24, page 13 "with the observed surface ridge structure", as well as in result and discussion section. It would be great to have more visual information (aerial image, geomorphological map, topography) to discuss these links.

Figure 2: all the other figures have similar range of values (colors) for ERT, except that one. For better comparison between the various figures, I suggest that the authors trim this color scale and clarify this in the caption.

Figure 6: there are some longitudinal features that can be clearly identified and discussed in the manuscript. It would be nice to see them in a fence diagram too (ERT transects reposition in space) (see major comment). This would also show the reader that these features are observed in multiple cross-sections.

---

## Author Comment (AC1) · 21 Oct 2016

**Author's response to Referee comments**

*RC = Comments from Referees*

AR = Author's response

**RM** = revised manuscript, changes are marked in **bold**
* * *
**Anonymous Referee #1**

**General Comments**

*RC.GC1*

*Even though a number of 2D geophysical surveys have been done on many rock glaciers, newly revealed subsurface heterogeneity is still valuable for discussing the kinematics and thermal history of rock glaciers. In particular, this paper shows detailed subsurface heterogeneity through a quasi-3D method. In addition to presenting valuable data, the authors well reviewed the related articles and carefully discussed the data. However, readers probably feel it difficult to follow the discussion on surface topography, because the visible information such as photos and maps are quite limited in this paper. To improve the paper, I recommend adding some geomorphological maps.*

AR.GC1

We agree and changed and enlarged the photos of figure 1 and added geomorphological maps (new figure 2) for a better site overview.

**RM.GC1**

**See new figures (fig. 1 and fig 2.) at the end of the document in original size.**
* * *
**Minor remarks**

*RC1.1*

*p.1 l. 25 topographical attributes and characteristics of the contributing area*

*It is difficult to understand what these terms mean.*

AR1.1

These terms summarize several heterogeneously distributed parameters. We changed the sentence and name these parameters now explicitly.

**RM1.1**

This is due to complex interactions between small-scale surface conditions **(e.g. grain size composition, snow cover distribution)**, topographical attributes **(e.g. aspect, slope)** and characteristics of the contributing area **(e.g. source of debris, extent of root zone)** (cf. Monnier et al., 2013; Luetschg et al., 2004; **Hanson and Hoelzle, 2004**; Harris and Pedersen, 1998)
* * *
*RC1.2*

*p.1 l. 27 to l. 2 (p. 2)*

*The sentences connecting with "Therefore" refer to different topics, thus the connection seems to be logically wrong.*

AR1.2

We agree and removed the logical connection.

**RM1.2**

[…] an enhanced knowledge of these interactions is needed. **The** detection and mapping of spatial variations within the internal structure can be seen […].
* * *
*RC1.3*

*p. 2 ll. 10-12*

*It would be better to revise the sentence, because the content of iii) is not equivalent to those of i) and ii). In general, iii) is firstly interpreted, then i) and ii) are discussed.*

AR1.3

We agree and revised the sentence.

**RM1.3**

We changed from:

For rockglaciers and similar periglacial landforms, geophysical investigations of the internal structure can e.g., i) reveal glacier-permafrost interactions during the development of rockglaciers (Dusik et al., 2015; Krainer et al., 2012; Ribolini et al., 2010), ii) provide inferences with creep velocities (Hausmann et al., 2012; Kneisel and Kääb, 2007) and iii) enable an assessment of subsurface material composition (Schneider et al., 2013; Musil et al., 2006).

To:

For rockglaciers and similar periglacial landforms, geophysical investigations of the internal structure **enable an assessment of the subsurface material composition and to distinguish between frozen and unfrozen areas (e.g. Schneider et al., 2013; Musil et al., 2006). This information can be used to investigate inferences with creep velocities (Hausmann et al., 2012; Kneisel and Kääb, 2007) or to reveal glacier-permafrost interactions during the development of rockglaciers (cf. Dusik et al., 2015; Krainer et al., 2012; Ribolini et al., 2010)**.
* * *
*RC1.4*

*p. 3*

*It would be better to add a comment whether direct observation of the borehole core is available or not. If available, it would be much better to discuss the composition of the rock glaciers with the core stratigraphy.*

*AR1.4*

We applied a destructive method, so no borehole cores are available. We added a comment.

**RM1.4**

**As a destructive drilling method was applied, no borehole cores are available.**
* * *
*RC1.5*

*p.3 l. 14 2D SRT*

*This is incorrect. They only did conventional refraction survey.*

AR1.5

The reviewer is right. We changed this information.

**RM1.5**

[…] who performed a **seismic refraction sounding** and one 2D ERT measurement […].
* * *
*RC1.6*

*p.3 ll. 21-31*

*Readers cannot check the topography mentioned in this part, because the photo and map in Fig. 1 are too small. Add a geomorphological map indicating not only the survey lines but also the lateral moraine, ice patches, rooting zone, ridges/furrows and LIA glacier extent.*

*AR1.6*

We agree and added geomorphological maps as suggested.

RM1.6

**See fig. 2 at the end of the document in original size.**
* * *
*RC1.7*

*p. 7 l. 13 first slices*

*Which is the first slice in the figure?*

AR1.7

We changed the reference from the quantifier to the indicated depth level of the slice which is displayed in the figure.

**RM1.7**

[...] the talus cone (cf. fig. **5**, **model slice 0.5–1m**) to […].
* * *
*RC1.8*

*p.7 l. 25 the centre of the rockglacier*

*It may confuse readers, because the center of the rock glacier does not corresponds to the center of Fig. 4 but the lower edge.*

AR1.8

We refer to the centre of the rockglacier on a median plane but still remain in the upslope part of the rockglacier with the description. To avoid misunderstandings, we rephrased this paragraph.

RM1.8

[…] direction and towards the **lateral** margin of **this upslope part of** the rockglacier (up to 4 m thickness), while it remains thin in the **central upslope** part of the rockglacier **near the transition zone** (2 m to 3 m thickness).
* * *
*RC1.9*

*p. 8 l. 21 reaches a depth of 7 m*

*Is it correct? Fig. 5 indicates that the deeper part (11-13 m) also has similar resistivity.*

AR1.9

We see that the descriptions in the text and the corresponding structures of (former) figure 5 are difficult. To clarify what parts of the model slices are addressed, we split this structure (like we did with former structure 4 (now structure 3), changed the nomenclatures and highlighted the broad extent of the structures.

**RM1.9**

The first of the three structures is **60** m long and reaches a depth of 7 m **in the upslope part (1a). In the downslope part (2b), the lower boundary could not be delimited within the model**.

**See fig. 6 at the end of the document in original size.**
* * *
*RC1.10*

*p.8 ll. 22-23 The layer above this structure is highly variable in thickness and ranges from 4 m in the upslope part of the model over 1 m in the middle part to 5 m in the downslope part of the model.*

*It is impossible to understand where "this structure" is, because the relatively high resistivity structure (>6 kohmm) indicated as (2) is covered with a thin low resistivity layer only seen in the slice of 0.5-1m.*

AR1.10

This remark refers to the same problem as the comment above. To make this section more understandable, we changed nomenclature and highlighted the described structures in the figure.

RM1.10

**See fig. 6 at the end of the document in original size.**
* * *
*RC1.11*

*l. 32 to l. 1 (p. 9)*

*Readers cannot confirm the relation between surface topography and resistivity from Figs. 5 and 6. Add a hillshade image with contour lines of the survey area.*

AR1.11

We think that the geomorphological maps (new figure 2) and the new figure 8 (which shows the topography) helps to illustrate the relation.

**RM1.11**

**See figs. 2 and 8 at the end of the document in original size**
* * *
*RC1.12*

*p. 10 ll. 29-30 This is caused by the lower pore space volume and hence the lower ground ice volume*

*This interpretation is questionable. Reconsider basic permafrost physics that frozen finer soil holds more unfrozen water in it near the melting point.*

AR1.12

We agree on this point and included this in the discussion.

RM1.12

This is caused **by the physical properties of fine materials which can store a relatively high amount of liquid water at temperatures close to the melting point or even at sub-zero temperatures, resulting in the observed low resistivity values (e.g. Schneider et al., 2013). Additionally**, the lower pore space volume **can generally lower the** ground ice volume which can develop by freezing of unconfined water (Scapozza et al., 2011; Vonder Mühll et al., 2000).
* * *
*RC1.13*

*p. 11 ll. 22-26*

*I cannot understand what the authors want to say.*

AR1.13

In this section, we discuss the formation and shape of the frozen structure of Nair rockglacier. We rephrased the section.

**RM1.13**

We changed the text from:

This represents a gradual decrease of ice content and is associated with a stepwise increase of ALT. The formation of this wedge-shaped structure can originate from meltwater of the embedded ice which drains into the rockglacier during summer and refreezes (Isaksen et al., 2000). The ability of the pebbly material to store water and to reduce the speed of runoff is likely to support this process

(Ikeda et al., 2008). It also explains the lower ice content at the lateral margin of the rockglacier which would also be less affected by meltwater flow.

To:

**In this part of the rockglacier, the ability of the pebbly material to store water and to reduce the speed of runoff (Ikeda et al., 2008) supports the refreezing of the meltwater from the embedded sedimentary ice. This results in a gradual decrease of ice content in a downslope direction and towards the lateral margin of the rockglacier, which is less affected by meltwater flow than the main part of the rockglacier. Thus, it can explain the observed stepwise increase of ALT and forms the wedge-shaped structure.**
* * *
*RC1.14*

*p. 12*

*The paragraph of the section 5.4 is too long. It would be better to revise it into several paragraphs.*

AR1.14

We divided this section into three parts.

**RM1.14**

We broke the paragraph at the following positions:

[…] advances during Holocene could have overridden the rockglacier. ¶

The occurrence of buried ice of […]

[…] was mapped in the Dolomites by Seppi et al. (2015)

**The observed pattern of frozen conditions** […]
* * *
*RC1.15*

*P12 ll. 4-6 , where … 1998)*

*It would be better to remove this part from this sentence, because the topic is not discussed here but after the glacier-permafrost interaction.*

AR1.15

We changed the position of this sentence to the section where we discuss the development of the ridges and furrow topography.

**RM1.15**

**The observed pattern of frozen conditions below the ridges and unfrozen conditions below the furrows contrasts the commonly known distribution pattern, which shows colder temperatures in the furrows due to topographic or microclimatic effects (Hoelzle 1999; Harris and Pedersen, 1998).** The formation of the characteristic ridge-furrow […].
* * *
**Anonymous Referee #2**

**Major Comments**

*RC2.1*

*The use of quasi-3D inversion is great and very valuable. However, I think its value cannot be really evaluated here because Figure 4 is not sufficient to analyze results. I suggest that the author also show a fence-type figure displaying various transects in 3D. This would help to see the variations in both depth and elevations. Also comparing results from a few transect inverted separately vs together would be useful to help the reader understand the value of quasi-3D ERI.*

AR2.1

We added an additional figure (fig. 8) with fence-type illustrations from selected transects of both inversion approaches: independently 2D and 3D. We also added a description of the differences between both methods to the discussion section ('Methodological aspects').

**RM2.1**

**See fig. 8 at the end of the document in original size.**

**A comparison between independently inverted 2D models and corresponding 2D slices of the 3D models (fig. 8) shows differences between the results of both approaches. This affects the modelled resistivity values as well as the depths of the detected structures. At Nair rockglacier e.g., resistivity values in the centre of the longitudinal profile (between the perpendicular profiles) are between 12 kΩm and 16 kΩm in the 2D model while they are between 8 kΩm and 12 kΩm in the 3D model. Greater differences are shown in the upslope part of the longitudinal profile of Uertsch_01 data set. Around Y = 90 m, the 3D model shows a high-resistivity structure which is displayed with much lower resistivity values by the independently inverted 2D data. The high-resistivity structure of Uertsch_02 data set (referred to as '4' in fig. 7), shows a shallow active layer in the results of the 2D approach, while it is obtained directly below the surface in the 3D model. Structures detected by the 3D approach are shaped smoother than the same structures detected by the 2D approach which is likely the result of diagonal filters and interpolation effects.**
* * *
*RC2.2*

*The description of the Figure 4 (5 and 6) and discussion in the manuscript is great (including the discussion of past studies). However, it is very hard to follow the discussion given the limited information in the figures. Topography is likely a key property in such analysis and has been measured (as visible in Figure 4) (I suggest that the authors add a sentence to explain how they measured it in more details). It would be great to have a map of the topography so that below structure could be more easily compared to surface variability (topography and micro-topography). Figure 4 shows that topography has been included in inversion but currently it is hard to observe and compare surface and subsurface features. Also a geomorphological map or any other information about the surface properties could significantly strengthen the paper.*

AR2.2

We measured topography using an RTK-GNSS system or assessed it in the field and corrected the data afterwards. We added a sentence about the topography measurement in the method section.

Micro-topography can be obtained from the new created figure 8 and a better overview is presented in the geomorphological maps (fig. 2) and the new photos (fig. 1).

**RM2.2**

Topography **(cf. fig. 8) was measured at each electrode using an RTK-GNSS system (Uertsch_02) or assessed in the field (Nair, Uertsch_01). The assessed topography data was corrected afterwards using digital elevation data. Topography** was incorporated into the collated [...]

**See figs. 1, 2 and 8 at the end of the document in original size.**
* * *
*RC2.3*

*I agree that the interpretation of thermistor data, EMI and SRT is generally difficult at temperature around 0 C. I think it would be very useful to see temperature profiles at other times of the year too for enhanced understanding of where the system is expected to be more dynamic. Please consider adding such a dataset in a figure.*

AR2.2

We added temperature profiles from exemplary days which represent minimum and maximum temperatures (mean over all sensors) and mean values for each sensor to the figures (panel c).

**RM2.3**

**See figs. 3 and 4 at the end of the document in original size.**
* * *
*RC2.4*

*There are several locations where wording could be possibly improved. E.g., "Its" or "their" on Line 6 (what does the "its" refer to)?; "and hence the performance"(Line 10) sounds awkward; "homogeneous decrease" or "constant decrease" (Line 12) ?; etc.*

AR2.4

We checked and changed the mentioned parts. Further corrections were made throughout the text.

**RM2.4**

Interactions between different formative processes are reflected in the internal structure of rockglaciers. **Therefore, the** detection **of subsurface conditions** can help to enhance our understanding of landform development. For an assessment of subsurface conditions, [...].

At Nair rockglacier, we discovered a gradual descent of the frost table in a downslope direction and a **constant** decrease of ice content which follows the observed surface topography
* * *
**Minor Comments:**

*RC2.5*

*L. 28 on page 2: Consider rewording last sentence.*

AR2.5

We changed the word order.

**RM2.5**

Changed from:

'Although two-dimensional quantitative assessments of ground ice content have already been presented (Pellet et al., 2016; Hausmann et al., 2012; Hauck et al., 2011), we presume that a qualitative approach is sufficient for interpreting the results regarding rockglacier development, and as a quantitative approach would require an additional spatial assessment of porosity.'

To:

**We present a qualitative assessment of ground ice content which we presume is sufficient for interpreting the results regarding rockglacier development. Quantitative assessments, as presented e.g. by Pellet et al., (2016), Hausmann et al., (2012) or Hauck et al., (2011), would additionally require a spatial assessment of porosity.**
* * *
*RC2.6*

*L.31 on page 2: "best investigated" in the world? Please be more specific*

AR2.6

We limit it to the European Alps although it might be true for the whole world. We specified this.

**RM2.6**

The nearby rockglaciers Murtèl and Muragl **probably** belong to the best investigated rockglaciers in the **European Alps**.
* * *
*RC2.7*

*L.2 on page 3: "boreholes" at each site? One at each site? Please clarify. I also suggest that authors name the two sites in this section.*

AR2.7

Yes, there is one borehole at each site. We name the two sites in this section now explicitly.

**RM2.7**

In September 2014, **one** borehole **with a depth of 10 m was** drilled **at Uertsch rockglacier and one borehole with the same depth was drilled at Nair rockglacier. Both boreholes are** instrumented with thermistor strings with 15 temperature sensors.
* * *
*RC2.8*

*L.21, page 4: what is the inter-electrode spacing (2m?)*

AR2.8

Inter-line electrode spacing is 2 m for profiles in X-direction and 3 m for profiles in Y-direction. We added this information in an additional sentence.

**RM2.8**

**Inter-line electrode spacings of 2 m and 3 m were used for profile lines in X- and Y-direction, respectively.**
* * *
*RC2.9*

L.28, page 4:  standard deviation larger than 5%.  Please provide information on how this standard deviation is measured, by the acquisition system alone? Or using reciprocal measurements?

AR2.9

This value is measured by the acquisition system and shows the deviation between the results of several reciprocal measurements of the same datum point. We added this to the method section.

**RM2.9**

**Each datum point was measured between two and four times and standard deviation between these reciprocal measurements was saved for quality checks.**
* * *
*RC2.10*

*L.30 page 4: "collated". Consider using "merged"*

AR2.10

We exchanged the words following the suggestion.

**RM2.10**

The 2D data sets were **merged** into one single 3D file […].
* * *
*RC2.11*

*L. 17, page 5: the shots were all located on the transects. Please discuss shortly why no shot have been performed outside the geophone interval (as often done in seismic refraction).*

AR2.11

Indeed, shot points outside the transects were performed. We pointed this out.

**RM2.11**

**Additionally, offset shots outside the profile lines were performed.**
* * *
*RC2.12*

*Section 4.1: I got lost in this section because I did not understand what was discussed first. I suggest that the author clearly state that this section compare single ERT transect and SRT. I think personally that this section should be placed after the quasi-3D inversion and be only related to comparison between seismic and ERT.*

AR2.12

We placed this section in front of the quasi-3D inversion because we think that the comparative approach provides information which is valuable for the interpretation of the 3D models. It would be difficult to investigate subsurface conditions from the solely resistivity-based 3D models. The comparative approach, in combination with the borehole data, confirms the permafrost occurrence which is used as prior information for the 3D models.

We added an introduction to this section to point this out.

**RM2.12**

**A comparative analysis of single ERT and SRT profiles provides initial two-dimensional information on the local subsurface characteristics. The information of this comparative approach is valuable for interpreting the quasi-3D ERI models which are solely based on resistivity data. Additionally presented data from borehole temperature loggers gives information on the ground thermal regime and verifies the permanently frozen state of the subsurface of the investigated rockglaciers.**
* * *
*RC2.13*

*Line 5, page 6: "last two sensors". At which depths? The sentence is confusing, although I agree that temperature close to 0 C means that comparison between various dataset is difficult, as well as the assessment of permafrost presence or not.*

AR2.13

Depths are 9 m and 10 m, we added the specific values. The sentence refers to the accuracy range of the sensors (±0.25 °C) which we mentioned in an earlier section. We modified this sentence by repeating this information.

**RM2.13**

This means, that the difference from the freezing point is below the accuracy range of the sensors **(±0.25 °C)**. Year-round temperature logging (cf. exemplary plots in fig. 3c), shows that values of daily mean temperatures are consistently negative below a depth of 3 m, but only the sensors **at 9 m and 10 m depth** show values that are consistently lower than -0.25 °C **and hence confirm permafrost conditions**
* * *
*RC2.14*

*Line 5 on page 10: "removed" using which criteria*

AR2.14

We removed datum points using the following criteria: (i) more than 5% deviation between reciprocal measurements and (ii) RMS error higher than 100% in a trial inversion (procedure

proposed by Loke (2014)). We included this information in the method section and refer to this in the addressed sentence (L5 P10).

**RM2.14**

The measured apparent resistivity data sets were quality checked and bad datum points (standard deviation **between reciprocal measurements** > 5%) were deleted manually. […] **This includes an elimination of datum points with an RMS errors of > 100% in a trial inversion**.
* * *
*RC2.15*

*Line 9 on page 10: 'cut-off' value. Please clarify process. Although it is hard to evaluate penetration depth without knowing the acquisition geometry (a and n spacings), it seems to me that the bottom part in Figure 3 is not necessarily real. Please clarify your thoughts about this zone in the manuscript.*

AR2.15

We performed the model resolution approach that we describe in the method section for an evaluation of investigation depth. For the data set which is presented in (former) figure 3, low model resolution values in the bottom part of the tomogram indeed indicate that this bottom part is not necessarily real. We already state this on page 7, line 2: 'However, it must be noted that data coverage is low in this deep part of both models'.

To point this out, we split this joint statement into separate sentences. We additionally modified the description of the model resolution approach in the method section.

**RM2.15**

Description of figure 3 modified:

**However, low model resolution values in the bottom part of the model (not shown) indicate that this lower boundary is not necessarily backed by the measured data**. […] **As for the ERT profile**, it must be noted that data coverage is low in this deep part of **the** model.

New part in method section:

**To investigate** model reliability, **a** resolution matrix approach (**Hilbich et al., 2009**; Wilkinson et al., 2006; **Stummer et al. 2004**) was performed on all data sets. This approach provides a measure **of the information content of the model cells. It enables an assessment of the** independence of the modelled resistivity values from neighbouring cells or inversion settings. **We also used the approach to evaluate investigation depth. In RES3DINV, the model resolution values are transferred into index values, which additionally take into account the model discretization (Loke, 2015). We followed the suggested index value of 10 as a lower limit to rate model cells as sufficiently resolved.**
* * *
*RC2.16*

*L.15, p.9: "could not rediscover" or "did not rediscover (or observe)". Please clarify the potential reason. Is it because environmental/climate change, thermal state during survey, difference in acquisition geometry?*

AR2.16

We modified this sentence. The variation is likely caused by differences in data acquisition, but an influence of atmospheric warming also seems conceivable.

**RM2.16**

**However**, they **observed an ALT of** 2.2 m to 2.4 m which **is lower than what we observed in our results. This is likely caused by differences in the methodological approach or in the acquisition geometry. However, the impact of atmospheric warming during the last decade is also conceivable.**
* * *
*RC2.17*

*L24, page 13 "with the observed surface ridge structure", as well as in result and discussion section.  It would be great to have more visual information (aerial image, geomorphological map, topography) to discuss these links.*

AR2.17

We changed the photos presented in figure 1 and added some geomorphological maps (new fig 2). Information on topography can be obtained from the newly created figure 8.

**RM2.17**

**See new fig. 1, 2 and 8 at the end of the document in original size.**
* * *
*RC2.18*

*Figure 2: all the other figures have similar range of values (colors) for ERT, except that one. For better comparison between the various figures, I suggest that the authors trim this color scale and clarify this in the caption.*

AR2.18

We adjusted the colour scale of this figure to the same range that are used in the other figures.

**RM2.18**

**See fig. 3 at the end of the document in original size.**
* * *
*RC2.19*

*Figure 6:  there are some longitudinal features that can be clearly identified and discussed in the manuscript.  It would be nice to see them in a fence diagram too (ERT transects reposition in space) (see major comment). This would also show the reader that these features are observed in multiple cross-sections.*

AR2.19

We present transects which include these features in the fence-diagram of Uertsch_02 (new fig. 08). The reader can now see that these features are present in both X- and in Y-profiles.

**RM2.19**

**See fig. 8 at the end of the document in original size.**

**Revised figures**

**Figure 1**

[Figure]

Fig. 1: Site overview and measurement setups. a) Photo Nair site b), c) Photos Uertsch site d) quasi-3D ERI setup Nair e) quasi-3D ERI setups Uertsch_01 (blue lines) and Uertsch_02 (red lines).

**Figure 2**

[Figure]

Fig. 2: Geomorphological setting. Panels a), b) Nair, c), d) Uertsch

[Figure]

**Fig. 3: Comparative analysis of (a) 2D ERT Profile, (b) 2D SRT Profile and (c) 1D Temperature/ Resistivity/ Velocity plots at Nair rockglacier.**

**Figure 4**

[Figure]

**Fig. 4: Comparative analysis of (a) 2D ERT Profile, (b) 2D SRT Profile and (c) 1D Temperature/ Resistivity/ Velocity plots at Uertsch rockglacier.**

**Figure 6**

[Figure]

Fig. 6: Quasi-3D ERI Model Uertsch_01. Numbers refer to structures described in the text. Selected depth slices.

**Figure 8**

[Figure]

**Fig. 8: Comparison between independently inverted 2D models (central row) and 3D models (lower row). Upper row shows model representation as blocks and position of the selected slices.**